# Influence of Dietary Phytase Inclusion Rates on Yolk Inositol Concentration, Hatchability, Chick Quality, and Early Growth Performance

**DOI:** 10.3390/ani13061000

**Published:** 2023-03-09

**Authors:** Carlos Alexandre Granghelli, Carrie Louise Walk, Gilson Alexandre Gomes, Tiago Tedeschi dos Santos, Paulo Henrique Pelissari, Brunna Garcia de Souza Leite, Fabricia Arruda Roque, Mário Henrique Scapin Lopes, Lúcio Francelino Araujo, Cristiane Soares da Silva Araujo

**Affiliations:** 1Department of Animal Science, University of Sao Paulo, Pirassununga 13635-900, SP, Brazil; 2DSM Nutritional Products, Heanor DE75 7SG, UK; 3AB Vista, Marlborough SN8 4AN, UK; 4Department of Animal Nutrition and Production, University of Sao Paulo, Pirassununga 13635-900, SP, Brazil

**Keywords:** broiler breeder, glycerol, mineral, superdosing, yolk sac

## Abstract

**Simple Summary:**

Appropriate broiler nutrition is essential to guarantee good body development and a uniform growth rate. However, broiler breeder nutrition is also key for progeny success because the development of embryos and chicks depends directly on the nutrients in the yolk. The physiological status of embryos is influenced by the nutritional status of the breeder hens, which is reported to have a significant impact on progeny quality, development, and hatchability. Although phytase levels in broiler production are already consensual, there are no studies reporting the effects of phytase supplementation for broiler breeders on their progeny. The aim of this study was to evaluate how different levels of phytase supplementation, including superdosing, influence the performance and growth of progeny. In addition, the extra phosphoric effects of using high phytase supplementation to overcome the anti-nutritional effects of phytate were studied.

**Abstract:**

The aim of this study was to determine the influence of dietary phytase in breeder hens on yolk nutrients, hatchability, chick quality, and growth rate of their progeny, and their subsequent performance to 42 d post-hatch when fed diets with the same phytase concentrations. Breeder hens (*n* = 216) were divided into 3 groups receiving nutrient-adequate diets with reduced calcium (Ca) and phosphorus (P) (by 0.16% and 0.15%, respectively), supplemented with either 500, 1500, or 4500 FTU/kg phytase from 27 to 50 weeks of age. Eggs were collected at 38 weeks of age and incubated. On the day of hatch, the chick quality and hatchability were determined, and 18 chicks/group were euthanized for yolk sac collection and the determination of inositol and glycerol concentrations. The remaining chicks were divided into three groups, receiving different diets with reduced Ca and P (by 0.16% and 0.15%, respectively), supplemented with 0, 500, or 1500 FTU/kg phytase to 42 d post-hatch. Increasing the phytase concentration in the breeder hen diet linearly (*p* < 0.05) increased the number of early embryo deaths and decreased the number of late deaths and pips. The inositol concentration in the yolk sac at day of hatch increased (quadratic; *p* < 0.05) as the phytase dose increased in the breeder hen diet. The breeder hen diet (*p* < 0.05) influenced the body weight (BW), feed intake (FI), and feed conversion ratio (FCR) up to 21 days of age. The supplementation of breeder hen diets with 1500 FTU/kg phytase increased the concentration of sodium (Na), magnesium (Mg), potassium (K), manganese (Mn), and zinc (Zn) in the yolk sac. The inclusion of phytase doses up to 4500 FTU/kg appeared to influence embryo mortality, chick feed intake, and BW gain to 21 days and the FCR throughout the entire production phase.

## 1. Introduction

Breeder hen nutrition has been reported to have an influence on progeny quality, hatchability, and early growth rates [1,2,3]. Thus, the development of embryos and chicks depends directly on the nutrients in the yolk, and their physiological status is influenced by the nutritional status of the breeder hens [4]. In this context, an improvement in the digestibility of feed ingredients could improve the capacity for the transfer of nutrients from the hen’s diet to the egg. The use of exogenous enzymes in poultry feed has beneficial effects by increasing the bioavailability of nutrients and digestibility as well as helping to eliminate several anti-nutritional factors [5,6].

The poultry industry has used phytase supplementation for decades as a nutritional strategy to make available additional nutrients, including phytate phosphorus, which has secondary advantages of reducing environmental pollution [7,8,9] and decreasing the cost of feed formulation. However, the extra phosphoric effects are progressively evident and desired [10,11]. These extra phosphoric effects have been associated with increased nutrient availability through the degradation of phytate to inositol [12].

Inositol plays an important role in many physiological processes, including lipid transport and the function of coenzyme Q10. Coenzyme Q10 is an important antioxidant that aids mitochondrial functions, and thus improves not only energy metabolism, but also the nutrient transfer to the embryo through the egg yolk [13]. Studies have already shown that the use of high doses of dietary phytase increases free inositol concentrations in the gizzard, and improves weight gain and feed conversion in broilers [14]. In addition, previous research has shown that myo-inositol administered orally improves the performance of broiler chicks [15,16].

Furthermore, previous studies evaluating increasing doses of phytase in diets for laying hens [17] and catfish [18] have reported a significant increase in the mineral concentration in the liver, plasma, and, ultimately, egg yolk. The authors of these studies hypothesized that the mineral availability was increased due to an almost complete destruction of phytate. The improved performance associated with phytase supplementation for both broilers and laying hens is clear [6,19,20,21,22]. However, further studies are necessary to demonstrate how the supplementation of phytase in the diet of broiler breeders can influence chick performance and quality. Therefore, the aim of this study was to determine the influence of different dietary phytase inclusion rates for breeders on yolk inositol concentration, hatchability, chick quality, and early growth performance in progeny.

## 2. Materials and Methods

The study was conducted at the Poultry Science Laboratory of the School of Veterinary Medicine and Animal Science at the University of Sao Paulo, Pirassununga, SP, Brazil. The animal experiment was approved, and the experimental procedure followed the Institutional Animal Care and Use Committee Guidelines of the University of Sao Paulo (CEUA n. 9196040614).

### 2.1. Broiler Breeder Trial

The birds were housed in a shed with a negative ventilation system to allow the control of temperature and humidity. The facility consisted of pens with a capacity for 4 birds each (density of 2250 cm^2^ per bird), containing a trough feeder, nipple drinker, nesting box, and wood shavings as bedding.

A total of 216 AP95 Aviagen broiler breeder hens were randomly distributed between 3 treatment groups consisting of 18 replicates of 4 birds each. The birds were reared in a commercial environment up to 20 weeks of age, then transferred to the experimental facility. The management, photostimulation, and feeding practices during the rearing, breeding, and production phases followed the recommendations in the genetics manual [23]. From 20 to 26 weeks of age, all breeder hens received a common diet during the adaptation period in the new environment. After this period, the hens were reallocated to ensure similar average weights of birds in each pen. According to the lineage manual [23], the hens were expected to reach 5% production at 25 weeks and reach peak laying by 31 weeks. The experimental diets were offered at the beginning of week 27 until the end of the experiment, at 50 weeks of age. In addition, the hens were weighed at the beginning and at the end of the study. Mash feed was served once a day and each replicate received the same feed allocation, adjusted monthly in line with the egg production of each experimental unit, following the genetic performance manual (Table 1) [23]. During the production phase, the birds experienced 16 h of light/day.

The experimental treatments consisted of a basal diet based on corn and soybean meal supplemented with graded levels of phytase (500, 1500, and 4500 FTU/kg), with corresponding reductions in calcium and available phosphorus (by 0.16% and 0.15%, respectively). The diets were formulated to provide similar nutrient profiles that met the recommendations proposed in the manual of nutritional specifications [24], regardless of the enzyme treatment. The basal diet, containing the same macronutrient and micronutrient formulations (Table 2), varied only in the concentration of phytase.

The phytase used was an enhanced *E. coli* phytase (Quantum Blue), provided by AB Vista Feed Ingredients (Marlborough, UK), with an expected activity of 5000 FTU/g. The experimental diets were analyzed for phytase enzyme activity, calcium, and total phosphorus. Phytase enzyme activity was analyzed via a colorimetric enzymatic method, as previously described [25]. The calcium and total phosphorus determination followed the official guidelines [26]. 

A total of 40 roosters, also AP95 Aviagen and the same age as the hens, were used for artificial insemination and were kept in the same environment as the broiler breeders. However, they received a common diet without phytase because the male effect was not being considered in this study. At 37 weeks of age, semen was manually collected from the roosters. A small amount of feather trimming was performed around the cloacal area to facilitate visualization and avoid contact with excreta or debris. A week later, at 38 weeks of age and when the hens were in the mid-production cycle, semen was collected from the roosters by abdominal massage. The semen was pooled from 3 roosters, and the broiler breeder hens were immediately inseminated with fresh semen using a dose of 0.5 mL per hen. The eggs were collected from day 3 until day 10 after insemination and temporarily placed in a holding room at 18 °C. Subsequently, the eggs were grouped by treatment and incubated using standard procedures. Any soiled, cracked, and/or deformed eggs were not incubated. Eggs that failed to hatch were broken to determine the fertility (the number of fertile eggs divided by the number of incubated eggs multiplied by 100), and embryonic mortality was then classified as early (1 to 7 d), intermediate (8 to 14 d), and late (15 to 21 d). Additionally, the egg hatchability (the number of chicks hatched divided by the number of fertile eggs multiplied by 100) was evaluated.

### 2.2. Progeny Trial

The chicks were housed in an experimental pen facility in a masonry shed with open sides bound by screens and equipped with fans and foggers for temperature control. The temperature and humidity of the experimental house were assessed throughout the trial. On the day of hatching, 18 chicks/treatment were euthanized for yolk sac collection. The yolk sacs were freeze-dried and analyzed for the inositol and glycerol content using the methods described by [27] for the blood samples, adjusting the parameters and the molecular weight of the two analyzed components. The concentration of the minerals calcium (Ca), phosphorus (P), sodium (Na), magnesium (Mg), potassium (K), copper (Cu), iron (Fe), manganese (Mn), and zinc (Zn) was determined by optical emission spectrometry with inductively coupled plasma [26].

The remaining chicks were divided by sex and allotted into a 50:50 ratio. For each treatment, the chicks were allocated with 100 × 120 cm surface floor pens equipped with nipple drinkers and a tubular feeder, with rice husk as bedding. Feed and water were provided ad libitum for the duration of the study. Heating was provided using an infrared heating lamp when necessary. The light program was set as specified by the recommendations of the genetics manual [28] as follows: 23 h light and 1 h darkness until the chicks were 3 days old and then 18 h light and 6 h darkness until slaughtering age.

A total of 648 mixed chicks were equally divided into the experimental treatments to create a 3 × 3 factorial (breeder hen diet × progeny diet) (Table 3). Each treatment was replicated 6 times (12 birds each, including 6 of each sex). The animals were fed from hatching to day 42. The progeny diets consisted of a corn and soybean meal-based diet supplemented with graded levels of phytase containing 0, 500, or 1500 FTU/kg (Quantum Blue, AB Vista, Marlborough, UK).

The diets were formulated to provide a similar nutrient profile that met the recommendations proposed in the nutrition specifications manual [29] regardless of the enzyme treatment. Ca and available P were reduced by 0.16% and 0.15%, respectively. The diets were fed in a two-phase feeding program: starter (1 to 21 d) and grower/finisher (22 to 42 d) (Table 4). In the groups where the effect of the maternal diet on the progeny was evaluated, the birds received a common basal progeny diet (without the inclusion of phytase) to verify these effects.

Birds and feed were weighed at 1, 7, 21 and 42 d of age to determine BW, FI and FCR (calculated from FI divided by BW gain) on a pen basis.. The incidence of mortality was recorded daily.

### 2.3. Statistical Analysis

The data were analyzed using the fit model platform in JMP Pro v. 14.0 [30]. Each replicate was considered to be an experimental unit in both the parent and progeny experiments. The experimental model included the breeder hen diet for the chick quality, with a statistical analysis performed on the data for fertility (%); hatchability (%); early, mid, and late deaths (%); pips (%); and inositol and glycerol concentrations in the yolk content (µmol/g). The hatchability data were transformed using Box–Cox transformations and the untransformed means were presented. The experimental model also included the breeder hen diet, progeny diet, and interactions for the progeny performance parameters, including the body weight (g/bird), feed intake (g/bird), and feed conversion rate (g/g). All model factors were considered to be nominal variables. When the model effects were significant at *p* < 0.05, the means were separated using linear and quadratic orthogonal contrast statements.

## 3. Results

There was no effect of diet (*p* > 0.10) on hen weight at week 27 or 50. The weight on day 0 (189 days old; 27 weeks) was 2.6 kg and 4.1 kg at week 50. Phytase activity in the breeder diets was 336, 1160, and 4610 FTU/kg for 500, 1500, or 4500 FTU/kg, respectively. The average phytase activities in the progeny diets were < 50, 360, and 1510 FTU/kg (initial) and < 50, 670, and 1310 FTU/kg (grower/finisher) for 0, 500, and 1500 FTU/kg, respectively. Hen mortality was 0%, and egg production and total eggs/hen/week were as expected according to the breed guidelines from 27 to 50 weeks [23].

There was no significant effect of the breeder diet on the percentage of fertile eggs or hatchability (Table 5). The percentage of early embryo death linearly (*p* < 0.05) increased as the phytase dose increased in the breeder diet whereas the percentage of late deaths and pips linearly (*p* < 0.05) decreased (Table 5).

The inositol content in the yolk sac was greater in the progeny from hens fed 4500 FTU/kg and the progeny from hens fed 500 FTU/kg, resulting in a quadratic (*p* < 0.05) influence of breeder diet on yolk sac inositol concentration (Table 5). There was no effect of breeder diet on the glycerol content of the yolk sac on the day of hatching.

The breeder hen diet influenced the progeny performance to a greater extent than the chick diet post-hatching. For example, the initial BW and the BW at day 7 post-hatching linearly increased (*p* < 0.05) as the phytase dose increased from 500 to 4500 FTU/kg in the breeder hen diet (Table 6). No significant effect of the breeder diet on the BW of broiler chicks at day 21 or 42 post-hatching was observed (Table 6). Conversely, the progeny diet supplementation with phytase had no significant effect on the BW throughout the production phase. The breeder diet affected the feed intake from 7 to 21 days post-hatch (*p* < 0.05), with a linear increase as the phytase dose increased from 500 to 4500 FTU/kg (Table 6). 

There was no significant effect on the FI during the phytase supplementation of the progeny diet from 1 to 42 days of age. The breeder diet continued to influence the FCR up to 21 days post-hatching (Table 6). There was a quadratic effect at 7 and 21 days of age (*p* < 0.02 and *p* < 0.01, respectively), and the FCR increased with the use of 4500 FTU/kg (Table 6). In addition, there was a significant quadratic effect of phytase supplementation on the progeny diet at 21 d, with an increase in the FCR when at a dose of 500 FTU/kg.

There was no interaction between the breeder diet and the progeny diet and the BW, FI, or FCR.

The supplementation of phytase in the breeder diet influenced the concentration of macrominerals in the yolk sac. There was a quadratic effect on the concentrations of Mg and K (*p* < 0.01) with an increase in the concentration of these minerals up to the dose of 1500 FTU/kg phytase (Table 7). There was a linear increase in the Na concentration (*p* < 0.05) with increasing doses of phytase in the maternal diet (Table 7). There was no significant effect on the Ca and P mineral concentrations (Table 7). Phytase supplementation in the maternal diet also influenced the concentrations of microminerals in the yolk sac. There was a significant and quadratic effect on the Mn concentration (*p* < 0.05), increasing up to the dose of 1500 FTU/kg. 

## 4. Discussion

There are numerous causes of embryo mortality, such as insufficient time of egg storage, incorrect incubation temperatures, inadequate humidity or ventilation, mechanical impact during transport, contamination, or nutritional deficiencies. Early embryonic mortality on average was 6.75%; this was slightly lower than that observed by [31], but greater than the guidelines at 3.5% [23]. In our study, there was a significant reduction in late embryonic mortality as well as pips in the egg shell with an increasing phytase supplementation in breeder diets. This may have been due to the increase in inositol, which can be used by the chick as a nutrient to complete the hatching process. The increased early embryonic mortality in the diets with increased doses of phytase in the breeder diet was not expected. Nonetheless, the increase in early deaths combined with the decrease in late deaths could indicate an increase in carbohydrate metabolism during the embryonic growth cycle of the chick, which could result in toxic byproducts such as CO_2_ or lactic acid very early in the embryonic stage before the establishment of adequate respiratory surfaces, as stated by [31]. 

The progeny performance was influenced by the breeder hen diet to a greater extent than the diet the chicks were fed post-hatching for the BW, FI, and FCR, with the post-hatching diet only significant for the FCR during the final 21 days of rearing (Table 6). Differing from the results found in the present study, a study evaluating two levels of phytic phosphorus (0.22% and 0.44%) and three concentrations of phytase (0, 500, and 1000 FTU/kg of feed) in broiler diets showed that phytase improved the weight gain of broilers [32], although there was no difference between the two levels of phytic phosphorus studied. In agreement with the results in the current experiment, a study using two diets supplemented with phytase (750 FTU/kg and 1500 FTU/kg of feed) showed no effect on the body weight gain of broilers [33]. 

The increase in the FI as the phytase dosages in the breeder diet increased may have been due to the intrinsic relationship between the initial BW and FI. Birds with a greater body development will have a better development of internal organs, especially the intestine. The growth rate is known to be partially mediated by the development of different organs [34]—namely, the duodenum—the weight of which increases as the BW gain increases [35]. The effects of the hen diet did not persist beyond 21 days of age. This difference may have been more noticeable in the initial feeding phase because broilers have a greater food voracity in the finishing stage. Corroborating the results of our study, no differences in the FI were observed when broilers diets were supplemented with phytase at 750 FTU/kg or 1500 FTU/kg [33]. 

The performance results obtained in this experiment contradicted the hypothesis that the performance of birds may continue to improve with increasing the phytase supplementation of diets above the standards recommended by the industry. The absence of effects on the BW and FI and the increase in the FCR at 42 d after the supplementation with higher doses of phytase in the broiler diet may have been due to an insufficient reduction in the levels of calcium and, more particularly, available phosphorus in the diet. The performance of chickens and the supplementation of phytase in the diet depends on the levels of non-phytic phosphorus [36]. According to these authors, a supplementation of 600 FTU/kg in broiler diets resulted in an increase in weight gain, FI, and tibia ash percentage and a reduction in mortality. The authors showed that phytase was able to reduce the requirements for non-phytic phosphorus by 8.5% for weight gain, 3.5% for FI, and 6.5% for the tibia ash percentage. The phytase effect was more evident in diets with low, rather than with high, concentrations of non-phytic phosphorus. However, another study showed that broilers consuming diets containing 12,500 FTU/kg of feed had a 6% greater weight gain and a 9.4% lower FCR when compared with broilers that consumed a positive control diet [37]. Therefore, improvements in broiler performance have been observed with higher phytase doses than those added to the progeny diet in the current study. 

In contrast, improvements in the BW (hatch day and 7 d), FI (7 and 21 d), and FCR (7 and 21 d) were observed in the progeny with a maternal supplementation of phytase, suggesting that it might be better to focus phytase supplementation on the breeder hen rather than broiler chickens, or that there may be benefits to re-evaluating the ratio between the supplementation of both diets. Therefore, further studies are needed to better understand the relationship between phytase supplementation for breeders and the effects on the progeny.

The improvements in the availability of Na, Mg, and K could be explained by the greater absorption of minerals due to the reduction in the concentration of phytates in the diets, as observed by others in other species [17,18,38]. In the present study, this was more evident in the diets containing 1500 and 4500 FTU/kg phytase. Ca and P concentrations varied in a manner different from the other macrominerals. This could be explained by the increasing demand for Ca and P in embryonic development during the early growth of the chick.

The result observed for Zn may have been the effect of the reduction in phytate in the diet due to the destruction of the molecule by greater phytase concentrations; hence, the availability of microminerals increased, as was also observed to a lesser extent in the Mn concentrations. The supplementation of diets with higher levels of phytase in the breeder diet did not influence the concentrations of the minerals Cu and Fe. Differences in the concentrations of the minerals Na, K, Mn, and Zn in the yolk sac indicated that phytate not only interfered with the absorption of Ca and P, but also had a much broader effect on the nutrition of broiler breeders. The authors, therefore, suggest that a higher mineral concentration in the yolk sac may have a direct effect on hatchability and initial chick development.

There have been few studies investigating the effects of breeder hen dietary phytase on progeny, and there remains a degree of uncertainty as to the dose that will best support broiler development. Arguelles-Ramos et al. [39] aimed to analyze the effects of phytase inclusion on broiler breeders, although this study did not focus on superdosing or progeny performance and development. Berry et al. [40] studied the supplementation of 300 FTU/kg phytase for breeder hens, analyzing the effect of phytase on the bone mineral content and density as well as the breeder performance, but without considering the progeny. Therefore, this is the first known report analyzing superdoses of phytase in broiler breeder hens, and following the effect on the hatchability, performance, and growth of progeny.

## 5. Conclusions

In summary, breeder hen nutrition influences progeny hatchability and early growth rates. Increasing phytase doses up to 4500 FTU/kg significantly increases the inositol concentration of the yolk sac, which may be used by the embryo as an energy source during hatching. This may decrease the number of late deaths during incubation and pips as well as increasing the chick body weight on the day of hatching. However, an increase in early embryo deaths was seen as the phytase doses increased. The supplementation of superdoses of phytase for breeder diets also resulted in an improvement in the BW, FI, and FCR up to 21 days post-hatching in chicks in addition to increasing the availability of Na, K, Mn, and Zn in the yolk sac on the day of hatching.

## Figures and Tables

**Table 1 animals-13-01000-t001:** Feed intake of breeder hens from 27 to 50 weeks of laying.

Feed Intake/Pen (g)						
Week 27–30	Week 31–34	Week 35–38	Week 39–42	Week 43–46	Week 47–50	Week 27–50
676.0	676.0	670.0	662.0	654.0	646.0	664.0

**Table 2 animals-13-01000-t002:** Ingredients and calculated composition in an experimental diet for broiler breeders fed superdoses of phytase.

Ingredients (%)	Basal Diet
Corn	63.70
Soybean meal	22.92
Wheat middlings	3.12
Soybean oil	1.44
Dicalcium phosphate	1.06
Vitamin premix ^1^	0.10
Mineral premix ^2^	0.10
Salt	0.41
DL-Methionine (99%)	0.11
L-Lysine HCL (78.4%)	0.06
Limestone	6.98
Total	100.00
Calculated Composition	
Metabolizable energy (kcal/kg)	2.82
Crude protein (%)	16.50
Calcium ^3^ (%)	3.04
Available phosphorus ^3^ (%)	0.30
Digestible lysine (%)	0.77
Digestible methionine + cysteine (%)	0.59
Digestible methionine (%)	0.58
Digestible threonine (%)	0.54
Digestible valine (%)	0.67
Linoleic acid (%)	2.20

^1^ Vitamin premix provided per kg of diet: vitamin A, 9000 UI; vitamin D3, 2600 UI; vitamin E, 14 UI; vitamin K3, 1.6 UI; vitamin B1, 2.2 mg; vitamin B2, 6 mg; vitamin B6, 3 mg; vitamin B12, 10 mcg; nicotinic acid, 0.03 g; pantothenic acid, 0.005 g; folic acid, 0.6 mg; biotin, 0.1 mg. ^2^ Mineral premix provided per kg of diet: Zn (ZnO), 0.126 g; Cu (CuSO_4_), 0.0126 g; I (Ca(IO_3_)_2_), 2.52 mg; Fe (FeSO_4_), 0.105 g; Mn (M_n_SO_4_), 0.126 g. ^3^ Matrix estimated by phytase: 0.16% Ca and 0.15% available P. The mean value analyzed was 3.18% and 0.50% for Ca and total P, respectively.

**Table 3 animals-13-01000-t003:** Progeny experiment design: completely randomized 3 × 3 factorial design.

Treatment	Breeder Diets	Progeny Diets
1	Basal diet + 500 FTU/kg	Basal progeny diet (BP)
2	Basal diet + 500 FTU/kg	BP + 500 FTU/kg
3	Basal diet + 500 FTU/kg	BP + 1500 FTU/kg
4	Basal diet + 1500 FTU/kg	BP
5	Basal diet + 1500 FTU/kg	BP + 500 FTU/kg
6	Basal diet + 1500 FTU/kg	BP + 1500 FTU/kg
7	Basal diet + 4500 FTU/kg	BP
8	Basal diet + 4500 FTU/kg	BP + 500 FTU/kg
9	Basal diet + 4500 FTU/kg	BP + 1500 FTU/kg

**Table 4 animals-13-01000-t004:** Progeny trial: composition of the basal progeny diets.

Ingredients (%)	Initial	Grower/Final
Corn	61.20	64.73
Soybean meal	34.53	29.53
Dicalcium phosphate	1.00	1.64
Soybean oil	1.20	2.19
Limestone	0.94	0.79
Salt	0.42	0.42
L-Lysine HCl (78.4%)	0.20	0.25
DL-Methionine (99%)	0.08	0.24
Vitamin premix ^1^	0.05	0.05
Mineral premix ^2^	0.10	0.10
Threonine	0.09	0.06
Total	100.00	100.00
Composition Calculated		
Metabolizable energy (kcal/kg)	2983	3100
Crude protein (%)	21.27	19.41
Calcium ^3^ (%)	0.74	0.82
Available phosphorus ^3^ (%)	0.30	0.41
Digestible methionine + cysteine (%)	0.87	0.77
Digestible lysine (%)	1.21	1.07
Digestible methionine (%)	0.47	0.77
Digestible threonine (%)	0.79	0.70

^1^ Vitamin premix provided per kg of diet: vitamin A, 6000 UI; vitamin D3, 2000 UI; vitamin E, 10 UI; vitamin K3, 1.6 UI; vitamin B1, 1.4 mg; vitamin B2, 4 mg; vitamin B6, 2 mg; vitamin B12, 10 mcg; niacin, 0.03 g; pantothenic acid, 0.011 g; folic acid, 0.6 mg. ^2^ Mineral premix provided per kg of diet: Zn (ZnO), 0.126 g; Cu (CuSO_4_), 0.0126 g; I (Ca(IO_3_)_2_), 2.52 mg; Fe (FeSO_4_), 0.105 g; Mn (M_n_SO_4_), 0.126 g. ^3^ Matrix estimated by phytase: 0.16% Ca and 0.15% available P. The mean value analyzed was 0.77% Ca and 0.53% total P (initial diet), and 0.78% Ca and 0.63% total P (grower/final diet).

**Table 5 animals-13-01000-t005:** Influence of phytase supplementation in 38-week-old breeder diets on progeny quality and yolk sac nutrient concentration on day of hatching.

	Breeder Diet (FTU/kg)					Contrasts ^1^	
Variable	500	1500	4500	SEM ^3^	*p*	L	Q
Fertility (%)	97.0	95.9	96.5	1.3	0.92	-	-
Hatchability (%)	81.5	88.0	84.6	3.1	0.40	-	-
Early death ^2^ (%)	3.44	6.01	10.8	1.7	0.02 ^4^	0.006 ^4^	0.61
Intermediate death ^2^ (%)	0.00	1.12	0.64	0.5	0.23	-	-
Late death ^2^ (%)	6.37	1.62	1.46	1.4	0.03 ^4^	0.02 ^4^	0.19
Pip (%)	7.46	2.32	1.91	1.9	0.08	0.04 ^4^	0.31
Inositol (µmol/g)	1.36	1.22	1.52	0.07	0.02 ^4^	0.13	0.02 ^4^
Glycerol (µmol/g)	1.58	1.48	1.50	8.17	0.66	-	-

^1^ L: linear contrast statement; Q: quadratic contrast statement. ^2^ Early death: incubation days 1–7; intermediate death: incubation days 8–14; late death: incubation days 15–21. ^3^ Standard error of the mean. ^4^ Results with significant (*p* < 0.05) values.

**Table 6 animals-13-01000-t006:** Influence of phytase dose in diets of 38-week-old breeder hens and the subsequent effect of phytase dose in the diets of the progeny from hatching to day 42.

Phytase (FTU/kg)	BW (g/bird)				FI (g/bird)			FCR (g/g)		
	1 d	7 d	21 d	42 d	7 d	21 d	42 d	7 d	21 d	42 d
Breeder Diet										
500	47.7	168	901	2858	126	1138	4713	1.05	1.33	1.70
1500	47.9	173	926	2872	131	1154	4753	1.04	1.31	1.68
4500	48.6	181	929	2862	146	1200	4830	1.08	1.37	1.71
SEM ^1^	0.29	2.07	12.3	39.3	1.96	16.9	48.3	0.01	0.01	0.01
Progeny Diet										
0	47.9	175	908	2848	133	1140	4693	1.04	1.32	1.68
500	48.3	172	910	2893	134	1169	4781	1.07	1.37	1.70
1500	48.0	175	938	2852	135	1182	4821	1.06	1.33	1.72
SEM ^1^	0.29	2.08	12.2	39.3	1.96	16.9	48.4	0.01	0.01	0.01
Probability										
Breeder diet	0.08	<0.01 ^2^	0.22	0.97	<0.01 ^2^	0.03 ^2^	0.24	0.02 ^2^	<0.01 ^2^	0.09
Linear	0.03 ^2^	<0.01 ^2^	-	-	<0.01 ^2^	0.01 ^2^	-	0.02 ^2^	0.06	0.21
Quadratic	0.53	0.70	-	-	0.03 ^2^	0.47	-	0.06	<0.01 ^2^	0.07
Progeny diet	0.70	0.39	0.17	0.66	0.70	0.22	0.18	0.31	0.03 ^2^	0.07
Linear	-	-	-	-	-	-	-	-	0.66	0.02 ^2^
Quadratic	-	-	-	-	-	-	-	-	0.01^2^	0.79
Interaction	0.39	0.60	0.68	0.62	0.84	0.91	0.51	0.39	0.80	0.85

^1^ Standard error of the mean. ^2^ Results with significant (*p* < 0.05) values.

**Table 7 animals-13-01000-t007:** Mineral concentration of yolks sac collected from one day-old chicks of broiler breeders fed superdoses of phytase from 27 to 50 weeks of age.

Phytase (FTU/kg)	Ca (%)	Na (%)	Mg (%)	K (%)	P (%)	Cu (ppm)	Fe (ppm)	Mn (ppm)	Zn (ppm)
500	0.975	0.116	0.036	0.101	0.534	7.39	51.7	3.28	38.4
1500	1.040	0.129	0.046	0.139	0.539	11.70	46.9	4.45	44.5
4500	1.105	0.136	0.036	0.118	0.507	18.83	46.4	2.98	48.9
SEM ^1^	0.078	0.006	0.002	0.006	0.015	5.24	3.1	0.37	3.2
Probability									
Breeder diet	0.49	0.05	<0.01	<0.01	0.28	0.29	0.40	0.02	0.06
Linear	-	0.01	0.83	0.05	-	-	-	0.58	0.02
Quadratic	-	0.66	<0.01	<0.01	-	-	-	<0.01	0.82

^1^ Standard error of the mean.

## Data Availability

The data presented in this study are available on request from the corresponding author.

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
