# Peer review of "Influence of Dietary Phytase Inclusion Rates on Yolk Inositol Concentration, Hatchability, Chick Quality, and Early Growth Performance"

_animals, 2023, doi:10.3390/ani13061000_

Round 1

Reviewer 1 Report

Under  statistical analysis, the experimental model used in the study should be specified and the variables in the model should be included indicating what each variable represents

Author Response

#Reviewer 1

Dear reviewer, we appreciate the feedback and comments. Changes were made as stated below:

L191, regarding the experimental models used

Experimental unit was clarified, as well as the variables included in the experimental model. The authors believe that the statistical analysis is much more comprehensive now for readers.

Reviewer 2 Report

The examples of studies on poultry, and other monogastric animals nutrition, testing  “superdoses” of phytase enzymes (i.e. inclusions higher than ca 500 FTU/kg feed) are very numerous in the literature. Moreover, in the past decade an extraphosphoric effects of phytases (e.g. upturns in digestive enzyme activity and energy/AA digestibility, diminishing of antinutritive effects of phytate and liberation of inositol), allowing not only for the reduction of dietary iP, but also increased growth and meat and/or egg yield, have become fairly well recognized.

In the above context, the purpose of this study/article (“to determine the influence of different doses of phytases [plural?] in broiler breeder diets [plural?] on hatchability, chick [?] quality, and yolk sac nutrient concentration of chicks at day of hatch, and determine the influence of phytase supplementation on breeder hen diets and phytase supplementation to the progeny on growth performance to 42-days post-hatch.”, line 75-79) is utterly incomprehensible. What is worse, both the title and the formulation of a research aim in the Abstract (line 26-28): “The objective of this trial was to determine the influence of a diet [?] on growth performance, hatchability, chick quality, and yolk [?] nutrient concentration of breeder hen chicks from day of hatch and subsequent starter [?] diet to 42 d post-hatch”, misinform readers about the intent of the study and the experimental design.

If the authors’ goal (as it can be guessed) was to assess the carry-over effects of supplementing broiler breeder (Ross 308 AP) hens between weeks 27 and 50 of age with different Phy doses (including two superdoses: 1,500 and 4,500 FTU/kg) on their mixed-sex progeny growth performance to 42 days of age, the phytase-unsupplemented “progeny” diets should only be used, and the “breeder trial” should include the negative control treatment (i.e. without phytase, just for the record).

After having carefully read the content, I conclude that this is a student’s paper accidentally co-authored by eminent professora. There are a number of deep concerns regarding the materials and the methods used. Many methodological details are either omitted, (e.g. environmental conditions during “progeny trial”) or unduly inflated (line 130-139: too many words, too much redundant information), or presented in unacceptable sparse form (e.g. inositol analysis - a core dependent variable: instead of an exhaustive description reference to published article, in which (BTW) analysis procedure of glycerol (a major contributor to gluconeogenesis during chick embryonic development) is not presented! [entry no 27 in the reference list]. On page 6 of this manuscript (subsection 2.3. Statistical analysis) it is claimed that “When model effects were significant at p < 0.05, means were separated using linear and quadratic orthogonal contrast statements. Trends were discussed at p < 0.10.”. In my opinion, analysing and presenting relatively small sample experimental data in a subjective manner by applying the term trend (= ”almost significant”) when P value is greater (0.07–0.09) than the pre-determined alpha (here < 0.05) demonstrates a misunderstanding of the meaning of P values. This, in my judgement, undermines claims of any statistical confirmation of differences between the treatment means by first or second-degree polynomial contrasts, as for example BW at day 1 and FCR at day 42 (Table 6) or yolk sac Na and Zn concentration values (Table 7).

Besides, citing findings from other studies cannot be a substitute for discussion of the results presented in the original articles. Considering the above, the discussion of the results reported in this manuscript actually does not exist. The fact that, according to the authors, “this is the first known report that analyzed super doses of phytase in broiler breeder hens and the following effect on the progeny hatchability, performance and growth” (line 333-335) is not a value in itself.    

For all of the foregoing reasons, I believe that this work is of low scientific quality such that it is not suitable for publication in a reputable journal.

Author Response

#Reviewer 2

Dear reviewer, we appreciate the comments made and thoughtful insights. Most of comments were added or further discussed.

The examples of studies on poultry, and other monogastric animals nutrition, testing  “superdoses” of phytase enzymes (i.e. inclusions higher than ca 500 FTU/kg feed) are very numerous in the literature. Moreover, in the past decade an extraphosphoric effects of phytases (e.g. upturns in digestive enzyme activity and energy/AA digestibility, diminishing of antinutritive effects of phytate and liberation of inositol), allowing not only for the reduction of dietary iP, but also increased growth and meat and/or egg yield, have become fairly well recognized.

The reviewer is correct when stating that there are examples of studies on poultry and monogastric regarding “superdoses” of phytase and extra phosphoric effects. However, as stated in the introduction section, (L58-67 and L76-81), the overall majority of those studies are regarding broiler chicken (especially) and laying hens (to a lesser extent), whereas the current study aimed to evaluate those effects on broiler breeders and how this supplementation in breeders can influence the following offspring, being a important topic to be studied.

In the above context, the purpose of this study/article (“to determine the influence of different doses of phytases [plural?] in broiler breeder diets [plural?] on hatchability, chick [?] quality, and yolk sac nutrient concentration of chicks at day of hatch, and determine the influence of phytase supplementation on breeder hen diets and phytase supplementation to the progeny on growth performance to 42-days post-hatch.”, line 75-79) is utterly incomprehensible.

The plural parts present in this statement have been corrected; Other mentions to phytase were already worded in singular. Authors appreciate the attention to detail. Proper phrasing was also corrected, as “to determine the influence of different dietary phytase inclusion rates for breeders on yolk inositol concentration, hatchability, chick quality and early growth performance”.

 What is worse, both the title and the formulation of a research aim in the Abstract (line 26-28): “The objective of this trial was to determine the influence of a diet [?] on growth performance, hatchability, chick quality, and yolk [?] nutrient concentration of breeder hen chicks from day of hatch and subsequent starter [?] diet to 42 d post-hatch”, misinform readers about the intent of the study and the experimental design.

Authors agree with the statement made by the reviewer, and another reviewer also pointed the same issue. It has been corrected and reworded to not misinform readers about the study objective.

If the authors’ goal (as it can be guessed) was to assess the carry-over effects of supplementing broiler breeder (Ross 308 AP) hens between weeks 27 and 50 of age with different Phy doses (including two superdoses: 1,500 and 4,500 FTU/kg) on their mixed-sex progeny growth performance to 42 days of age, the phytase-unsupplemented “progeny” diets should only be used, and the “breeder trial” should include the negative control treatment (i.e. without phytase, just for the record).

Authors believe that the “the carry-over effects of supplementing broiler breeder (Ross 308 AP) hens between weeks 27 and 50 of age with different Phy doses (including two superdoses: 1,500 and 4,500 FTU/kg) on their mixed-sex progeny growth performance to 42 days of age” can be evaluated despite not all progeny diets being phytase-unsupplemented, since the evaluation of further supplementation on the progeny diet can also be assessed and compared to only breeder supplemented diets.

As of the Breeder Trial not having a “Control treatment”, it was justified by the authors during the conceptualization of the study that the supplementation of 500 FTU/kg Phytase is already consolidated and considered a standard by the poultry industry, having no point in having a treatment without phytase since it is not viable for either birds or farmers. Thus, the 500 FTU/kg group is considered the “basal” treatment, and the super doses the ones being in fact evaluated in the study.

After having carefully read the content, I conclude that this is a student’s paper accidentally co-authored by eminent professora. There are a number of deep concerns regarding the materials and the methods used. Many methodological details are either omitted, (e.g. environmental conditions during “progeny trial”) or unduly inflated (line 130-139: too many words, too much redundant information), or presented in unacceptable sparse form (e.g. inositol analysis - a core dependent variable: instead of an exhaustive description reference to published article, in which (BTW) analysis procedure of glycerol (a major contributor to gluconeogenesis during chick embryonic development) is not presented! [entry no 27 in the reference list]. On page 6 of this manuscript (subsection 2.3. Statistical analysis) it is claimed that “When model effects were significant at p < 0.05, means were separated using linear and quadratic orthogonal contrast statements. Trends were discussed at p < 0.10.”. In my opinion, analysing and presenting relatively small sample experimental data in a subjective manner by applying the term trend (= ”almost significant”) when P value is greater (0.07–0.09) than the pre-determined alpha (here < 0.05) demonstrates a misunderstanding of the meaning of P values. This, in my judgement, undermines claims of any statistical confirmation of differences between the treatment means by first or second-degree polynomial contrasts, as for example BW at day 1 and FCR at day 42 (Table 6) or yolk sac Na and Zn concentration values (Table 7).

Authors do not agree that environmental conditions were omitted, since all details concerning the experimental shed, as well as pens, heating, humidity, and temperature control are present in the manuscript.

Authors agree with the reviewer’s statement about the glycerol analysis and have reworded the paragraph so it become clearer to readers. “Yolk sacs were freeze dried and analyzed for inositol and glycerol content, likewise the methods described by [27] for blood samples, adjusting parameters and the molecular weight of the two analyzed components.”

Authors agree that the wording regarding the abdominal massaging method may have been redundant, and thus was corrected. We would like to thank the reviewer for pointing it out, since it clarifies the reading.

Authors agree that the use of trends in this study was being used in such incorrect manner. Information regarding the discussion of “trends” was removed from the manuscript.

Besides, citing findings from other studies cannot be a substitute for discussion of the results presented in the original articles. Considering the above, the discussion of the results reported in this manuscript actually does not exist.

Authors find that stating that “The discussion of the results does not exist” an inappropriate phrasing regarding such academic work, and therefore do not agree with the statement.

This section discussed:

Possible causes of early deaths observed in embryos, as well as possible causes to the decrease of late death embryos; Comparisons between relevant studies that corroborate or not with the findings of the present study regarding broiler performance, reenforcing the available knowledge of the subject; Possible causes of the observed increase of FI as phytase doses increase; Performance results and calcium-phosphorus levels and reductions; Mineral availability and relationship with embryonic development and phytate concentrations.

Authors are aware and agree that the statement on lines 333-335 “is not a value in itself”. This phrase has been stated to inform readers about the poor literature available that causes investigations in this area to be mostly unclear. 

Authors would like to once again thank the reviewer for the time taken to review and evaluate the manuscript.

Reviewer 3 Report

As the authors identify the research subject had received limited attention and as the results suggest the approach to breeder hen feeding and management is relevant to progeny outcomes.

Review: Influence of different phytase doses on yolk inositol concentration, hatchability, chick quality and early chick growth rate of breeder hens

Heading Change to improve clarity

Influence of dietary phytase inclusion rates on yolk inositol concentration, hatchability, chick quality and early growth performance.

Ln 21: remove ‘behaves’ add ‘has’

Ln 23: remove ‘investigating the extra phosphoric effects of high supplementation of phytase to overcome the phytate’s antinutritional effects, and therefore promoting a significative improve on broiler nutrition and overall poultry industry.

Add ‘and investigate the extra phosphoric effects of using high phytase supplementation rates to overcome the phytate’s antinutritional effects, and therefore promote improved broiler nutrition.

Ln 26: remove ‘The objective of this trial was to determine the influence of a diet on growth performance, hatchability, chick quality, and yolk nutrient concentration of breeder hen chicks from day of hatch and subsequent starter diet to 42 d post-hatch.’

This lacks clarity       

Add ‘The aim was to determine the influence that breeder hen dietary phytase had on growth performance, hatchability, chick quality, and yolk nutrients of the chick progeny and their subsequent performance to 42 d post-hatch when fed  diets with the same phytase inclusion rates.’

Ln 36: remove ‘embryos early deaths’ add ‘early death embryos’

Ln 38: ‘Breeder hen diet (p < 0.05) influenced body weight (BW), feed intake (FI) and feed conversion ratio (FCR).

Comment: The authors should include that these improvements are until day 21 of age

Introduction

Ln 45: remove ‘of breeder’ add ‘of the breeder’

Ln 50: remove ‘maternal’ add ‘hen’s’

Ln 55: remove ‘due to an increase in the availability’ add ‘by increasing the availability’

Ln 60: remove ‘such as’ add ‘including’

Ln 64: remove ‘resulted in increased’ add ‘increase’

Ln 66: remove ‘the myo-inositol administered orally has indicated the improvement in performance of broiler chicks’ add ‘that myo-inositol administered orally improved the performance of broiler chicks’

Materials and Methods

Ln 135: remove ‘since hens’ add ‘when hens’

Results

In table 5 & 6: it would  be beneficial to include superscripts identifying where the differences are for those measures the at are significant.

Ln 222: remove ‘in’ add ‘to’

Ln 228: remove ‘in a linear raise’ add ‘as a linear increase’

Ln 232 and 233: remove ‘old’ add ‘of age’

Ln 236: remove ‘is reported’

Ln 237: remove ‘in the’ add ‘for’

Ln 241: insert space to start new sentence.

Ln 270: remove ‘FCR in average, being the post-hatch diet only significant on FCR during the last 21 days of age’ add ‘FCR, with the post-hatch diet only significant for FCR during the last 21 days of age’;

Ln 279: ‘The increase of FI as phytase dosages in breeder diet increase, as previously reported, may be due to the intrinsic relationship between initial BW and FI of birds with greater body development that have better development of internal organs, especially the intestine, and consequently have a higher FI.

Comment: This sentence is cumbersome and should be modified to improve the points being made. Also provide the reference for ‘as previously reported’

Ln 285; remove ‘a study used two diets supplemented with phytase (750 FTU/kg; 1,500 FTU/kg of feed), but found no differences in FI for broilers [33]. Add “no differences in FI were observed when broilers diets were supplemented with phytase at 750 FTU/kg or 1,500 FTU/kg of feed [33].

Ln 291; remove ‘in higher’ add ‘with higher’

Ln 292: remove ‘mainly’ add ‘and in particular’

Ln 298: remove ‘mainlyadd ‘rather than’

Ln 302: remove ‘Therefore, an improvement in broiler performance could be observed if higher doses of phytase were added to the progeny diet.

Add ‘Therefore, improvements in broiler performance have been observed at rates if higher than those added to the progeny diet in the current study.’

Ln 312: remove ‘in others’ add by others’

Ln 317: ‘The result observed for Mn and Zn may be the effect of the reduction of phytate in the diet due to the destruction of the molecule by greater phytase concentrations, hence the increased the availability of microminerals, similarly to macrominerals.

Comment: This sentence is cumbersome. While it appears Zn increases in linearly the effect on Mn is quadratic. So ‘by greater phytase concentrations’ holds for Zn but bot Mn

Ln 327: remove ‘There are few studies in this area, making it an uncertain field of research that has yet to be cleared and thus further studies need to be made, including which dose(s) may be the most beneficial to feed hens to provide proper broiler development’

Add ‘There have been few studies investigating effects of breeder hen dietary phytase on progeny and there remains a degree of uncertainty as to the most beneficial dose to support best broiler development.

Conclusion

Ln 340: ‘decreasing the number of late deaths during incubation and pips’

While this is the positive outcome the authors should not ignore the effect on early deaths as part of the conclusions at the very high phytase inclusion

Author Response

#Reviewer 3

Dear reviewer, we appreciate the review, suggestions and constructive criticism of our manuscript. Revisions regarding terms and/or spelling were exchanged and an English review was made. Authors would also like to thank the reviewer for pointing out suitable options to be added.

Heading

The title of the work has been remodeled as suggested, being less repetitive and more suitable: “Influence of dietary phytase inclusion rates of yolk inositol concentration, hatchability, chick quality and early growth performance.”

L21 Remove “Behaves”

Authors do not fully agree to this substitution, but understand how “behaves” is not the proper term to reference the effect on the following progeny.

L23 Remove “investigating the extra phosphoric effects of high supplementation of phytase to overcome the phytate’s antinutritional effects, and therefore promoting a significative improve on broiler nutrition and overall poultry industry.”

Authors appreciate this change, as it becomes clearer to understand the main goal of the study. Proper substitution has been made.

Ln 26: remove ‘The objective of this trial was to determine the influence of a diet on growth performance, hatchability, chick quality, and yolk nutrient concentration of breeder hen chicks from day of hatch and subsequent starter diet to 42 d post-hatch.’

The changes were made, since the new phrasing focuses on the breeder hen diet effects on the progeny, and not the variables analyzed in fact. Also, reworded in a way that keeps in line with the heading, as stated by reviewer #4

Ln 36: remove ‘embryos early deaths’ add ‘early death embryos’

Thanks for the correction. It has been reworded for proper understanding.

Ln 38: ‘Breeder hen diet (p < 0.05) influenced body weight (BW), feed intake (FI) and feed conversion ratio (FCR).

The comment has been added, since it reenforces the findings of the study during the early life of the chick.

Ln 45: remove ‘of breeder’ add ‘of the breeder’

Grammatical change has been made, focusing the nutrition per se

Ln 50: remove ‘maternal’ add ‘hen’s’

It is indeed more proper to address the breeder’s diets as the “hen’s” rather than “Maternal”;

Ln 55: remove ‘due to an increase in the availability’ add ‘by increasing the availability’

 Grammatical change has been corrected.

Ln 60: remove ‘such as’ add ‘including’

Example citing has been corrected to proper terminology.

Ln 64: remove ‘resulted in increased’ add ‘increase’

The comment was considered, since the stated effect was a fact and not a observation.

Ln 66: remove ‘the myo-inositol administered orally has indicated the improvement in performance of broiler chicks’ add ‘that myo-inositol administered orally improved the performance of broiler chicks’

The substitution was made to focus on the improvement effect observed in the study.

Ln 135: remove ‘since hens’ add ‘when hens’

Corrected, since it is clearer to state the timestamp rather than justifying the method.

In table 5 & 6: it would be beneficial to include superscripts identifying where the differences are for those measures the at are significant.]

Superscripts have been added to the respective tables, since it becomes more intuitive for readers and easier to make comparisons between diets.

Ln 222: remove ‘in’ add ‘to’

Grammar suggestions have been corrected and the authors appreciate the attention to detail.

Ln 228: remove ‘in a linear raise’ add ‘as a linear increase’

Rephrasing was accepted by the authors in order to keep clarity and formal writing.

Ln 232 and 233: remove ‘old’ add ‘of age’

Grammar suggestions have been corrected and the authors appreciate the attention to detail.

Ln 236: remove ‘is reported’

Unnecessary wording was detected by the reviewer. It was corrected.

Ln 270: remove ‘FCR in average, being the post-hatch diet only significant on FCR during the last 21 days of age’ add ‘FCR, with the post-hatch diet only significant for FCR during the last 21 days of age’;

Grammar suggestions have been corrected and the authors appreciate the attention to detail.

Ln 279: ‘The increase of FI as phytase dosages in breeder diet increase, as previously reported, may be due to the intrinsic relationship between initial BW and FI of birds with greater body development that have better development of internal organs, especially the intestine, and consequently have a higher FI.

Comment: This sentence is cumbersome and should be modified to improve the points being made. Also provide the reference for ‘as previously reported’

Authors agree with the reviewer’s statement and have reworded the paragraph so it become clearer to readers what is being discussed. Also, further references have been provided.

“The increase of FI as phytase dosages in breeder diet increase, as observed, may be due to the intrinsic relationship between initial BW and FI of birds with greater body development that have better development of internal organs, especially the intestine. Growth rate is known to be partially mediated by the development of of different organs (Lilja, 1983), naming the duodenum, being its weight known to increase as BWG increases (Wijtten et al., 2010). Therefore, birds with greater body development have better internal organ development, which may be a possible cause of having an increased FI.”

Ln 285; remove ‘a study used two diets supplemented with phytase (750 FTU/kg; 1,500 FTU/kg of feed), but found no differences in FI for broilers [33]. Add “no differences in FI were observed when broilers diets were supplemented with phytase at 750 FTU/kg or 1,500 FTU/kg of feed [33].

Better phrasing regarding the discussion step, changes were accepted by authors.

Ln 291; remove ‘in higher’ add ‘with higher’

Grammar suggestions have been corrected and the authors appreciate the attention to detail.

Ln 292: remove ‘mainly’ add ‘and in particular’

Better wording helps reader’s understanding, being corrected by authors.

Ln 302: remove ‘Therefore, an improvement in broiler performance could be observed if higher doses of phytase were added to the progeny diet.

Changes were accepted, since the conceptualization is in agreement with the findings.

Ln 312: remove ‘in others’ add by others’

Grammar suggestions have been corrected and the authors appreciate the attention to detail.

Ln 317: ‘The result observed for Mn and Zn may be the effect of the reduction of phytate in the diet due to the destruction of the molecule by greater phytase concentrations, hence the increased the availability of microminerals, similarly to macrominerals.

Comment: This sentence is cumbersome. While it appears Zn increases in linearly the effect on Mn is quadratic. So ‘by greater phytase concentrations’ holds for Zn but bot Mn

Authors agree with the statement, rewording the paragraph so it focusses on the linear effect on Zn, making an observation concerning the quadratic effect on Mn.

Ln 327: remove ‘There are few studies in this area, making it an uncertain field of research that has yet to be cleared and thus further studies need to be made, including which dose(s) may be the most beneficial to feed hens to provide proper broiler development’

Add ‘There have been few studies investigating effects of breeder hen dietary phytase on progeny and there remains a degree of uncertainty as to the most beneficial dose to support best broiler development.

Better phrasing was taken into account by authors, since it has become clearer about the current status of literature.

Ln 340: ‘decreasing the number of late deaths during incubation and pips’

While this is the positive outcome the authors should not ignore the effect on early deaths as part of the conclusions at the very high phytase inclusion

This is correct. Authors have now stated in “Conclusion” the unexpected results regarding the increase of early death embryos.

Reviewer 4 Report

Line 26: Corresponding to the title, need to make sure that the prime objective of this study is growth rate or yolk inositol concentration. Please review the sequence of objectives.

Line 41: Keeping in view the objectives, provide 1 line concluding statement in the abstract that which diet authors would like to recommend

Line 36:  Raised the number of embryo early deaths by 6.75%, should it be a concern?

Keeping in view the promising results of Phytase 4500FTU /Kg, what would be the tentative financial tentative of a commercial farm? How much it would be cost-effective?

Author Response

#Reviewer 4

Dear reviewer, we appreciate the comments and attention to points that were not clear to readers. Changes were accepted and corrected by authors

L26 Corresponding to the title, need to make sure that the prime objective of this study is growth rate or yolk inositol concentration. Please review the sequence of objectives.

The aim of the study was reworded to be in agreement with the heading provided, being the primary objectives stated first. Authors appreciate the suggestion.

Line 41: Keeping in view the objectives, provide 1 line concluding statement in the abstract that which diet authors would like to recommend

Authors agree and appreciate the suggestion. It is essential to have some type of concluding statement in the abstract section that resumes the results and conclusions of the study, and it has been added to the manuscript.

Line 36:  Raised the number of embryo early deaths by 6.75%, should it be a concern?

The raised number of embryo Early death is surely a concerned and the result was not expected by authors. Disclosures are present in the discussion section. Authors agreed that such discussion shouldn’t be added in the abstract section.

Keeping in view the promising results of Phytase 4500FTU /Kg, what would be the tentative financial tentative of a commercial farm? How much it would be cost-effective?

Historically, the cost of ingredients (representing cost of feed) has increased, while cost of phytase has decreased. This movement resulted in higher doses of phytase being commercially viable in animal production. Originally, inclusion rate of phytase would vary between 300 and 500FTU/kg while nowadays it is not uncommon to see the same product being added at doses of 1500 to 2500FTU/kg. Following the same trend in the future, it is not unexpected to even higher doses such as 4500FTU to be commercially viable to farmers.

Round 2

Reviewer 2 Report

The submitted text (animals-2206182-revised 1) represents pseudo-revised version of the original manuscript with a few cosmetic changes and additions made with “fine” words, e.g. “…there remains a degree of uncertainty as to…” (Line 334).

Both formulations of a research aim (Abstract: “…and growth rate of their progeny, and their subsequent  performance to 42 d post-hatch when fed diets with the same phytase concentrations.”; Line 76–78: “…and early growth performance in progeny.”) continue to mislead readers about the dietary treatments, monitored/calculated performance parameters, and the overall experimental design.

Besides, no reasonable justification has been provided for using the phytase-supplemented “progeny” diets (my key objection), all the more that broilers fed phytase additions showed significantly deteriorated feed conversion efficiency after 3 and 6 weeks of feeding.

Unfortunately, totally false claim on page 4 that yolk glycerol content was determined using the method described by Sophie A. Lee et al. (2018) [entry no 27 in the reference list] has not been eliminated and/or replaced with adequate reference, in spite of my clear remark that analysis procedure of glycerol is not described in this article. Therefore, I cannot see it as anything other than a fabrication of data on glycerol to make the manuscript “scientifically more attractive”. And in broad terms, this undermines the credibility of all presented findings.

I still believe that there is no real discussion of the results reported in this manuscript. For example, making surmises as to possible causes of early deaths observed in embryos (line 267–273) with reference to the user’s guide for JMP Pro v. 14.0 statistical package [entry no 30 in the reference list] seems like a joke.

I maintain my previous view that because of a low scientific quality this work is not suitable for publication in a reputable journal.